# Cardiac effects of OPA1 protein promotion in a transgenic animal model

Kitti Bruszt[1,2], Orsolya Horvath[1,2], Katalin Ordog[1,2], Szilard Toth[1,2], Kata Juhasz[3], Eszter Vamos[3], Katalin Fekete[3], Ferenc Gallyas[2,3], Kalman Toth[1,2], Robert Halmosi[1,2], Laszlo Deres[1,2]*

1 1st Department of Medicine, University of Pecs Medical School, Pecs, Hungary, 2 Szentagothai Research Centre, University of Pecs, Pecs, Hungary, 3 Department of Biochemistry and Medical Chemistry, University of Pecs Medical School, Pecs, Hungary

* deres.laszlo@pte.hu

**Data Availability Statement:** All relevant data are within the paper and Supporting information files.

**Funding:** The research in Hungary was funded by NKFIH within the framework of the University of Pécs' project TKP2021-EGA-17.

## Abstract

Mitochondria form a dynamic network in cells, regulated by the balance between mitochondrial fusion and fission. The inhibition of mitochondrial fission can have positive effects in acute ischemic/reperfusion injury models by preventing the fall in mitochondrial membrane potential associated with fission processes. However, inhibition of fission in chronic models is disadvantageous because it obstructs the elimination of damaged mitochondrial fragments. OPA1, in view of previous results, is a possible therapeutic target as a fusion promoter and structure stabilizer protein. We used transgenic mice in which the OMA1 cleavage sites of OPA1 were deleted. This resulted in a higher representation of L-OPA1 compared to S-OPA1. After genotyping and model validation, all animals were examined by echocardiograph on two occasions, at weeks 11 and 36. Histological samples were taken from hearts to examine mitochondrial morphology and structure remodeling. The signaling pathways related to mitochondrial dynamic processes were evaluated. Cardiomyocytes were isolated from neonatal mice to determine the efficiency of mitochondrial respiration using the SeaHorse assay method. OPA1 protein promotion has a negative effect on systolic function during aging. We confirmed that volume overload and ventricular remodeling did not manifest. The reason behind the loss of pump function might be, at least partly, due to the energy deficit caused by mitochondrial respiratory failure and damage in mitochondrial quality control pathways.

## Introduction

Mitochondria are the main source of cellular energy production through oxidative phosphorylation, accounting for 95% of the ATP synthesis [1, 2]. The morphology of the mitochondrial network is complex and highly variable due to mitochondrial dynamics, adapting to the cell's energy needs. Mitochondrial dynamics, involving fusion and fission processes, are crucial for cellular energy production, maintaining cellular health, regulating apoptosis, and ensuring the removal of damaged mitochondrial components. The dynamic balance influences spatial

**Competing interests:** I have read the journal's policy and the authors of this manuscript have the following competing interests: one of the coauthors, Ferenc Gallyas is a section editor of the journal.

distribution of mitochondria within cells, prevents disease-related imbalances, regulates reactive oxygen species formation as well as cellular signaling pathways. Overall, mitochondrial dynamics play a central role in sustaining cellular function and responding to changing physiological conditions [1–3]. The balance between fusion and fission processes is crucial for maintaining proper spatial distribution and optimal mitochondrial function [4–6].

Cytosolic proteins such as dynamin-related protein-1 (DRP1) and outer mitochondrial membrane partners, including fission 1 (Fis1), mitochondrial fission factor (MFF) and mitochondrial dynamics protein (49 kDa—MiD49 and 51 kDa—MiD51) control the fission process. Mitofusin 1 and 2 (Mfn 1 and 2) facilitate outer membrane fusion, while the long form of optic atrophy-1 (OPA1), a member of the GTPase family, promotes inner membrane fusion. Imbalances in the fission-fusion machinery can lead to impaired mitochondrial function, increased free radical production, opening of anion channels, and membrane depolarization, ultimately inducing mitochondrial fission [7–10]. To address this, damaged mitochondria are targeted for elimination through mitophagy, a process mediated by the PINK1/Parkin pathway [11, 12] (Fig 1).

Mitochondrial dynamics, including processes like fusion and fission, are implicated in cardiovascular diseases. Dysregulation of these processes, as we mentioned above, could cause disturbances in mitochondrial functions that play role in the development and progression of various cardiovascular conditions, including ischemic heart disease, cardiomyopathies, and heart failure [7–10]. Understanding and targeting mitochondrial dynamics may offer opportunities for therapeutic interventions in cardiovascular diseases. Therefore, alongside the investigation of compounds targeting the direct mitochondrial attack points (ie. MitoCsA, Cyclosporin), studies have emerged focusing on influencing mitochondrial dynamics, too.

# MITOCHONDRIAL QUALITY CONTROL

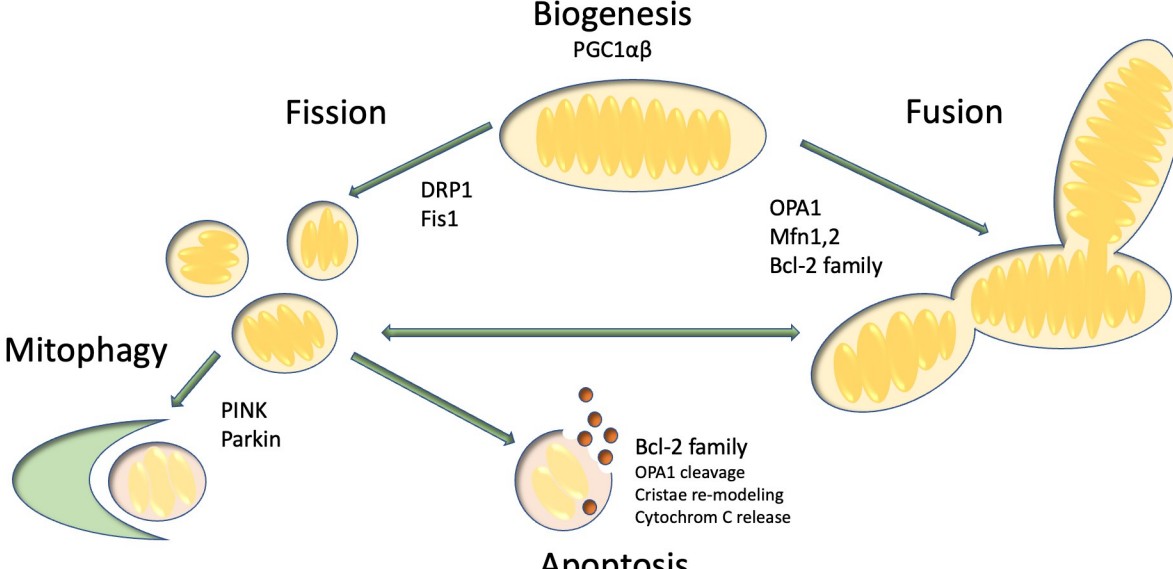

**Fig 1. Schematic picture of mitochondrial quality control processes.** Mitochondrial quality control consists of biogenesis, fusion, fission and mitophagy. To maintain optimal mitochondrial function, these processes are in dynamic equilibrium. Peroxisome proliferator-activated receptor gamma coactivator 1-alpha and beta (PGC1 α and β) regulate mitochondrial biogenesis. Mitochondrial fusion promoted by OPA1 and Mfn 1,2. Cleavage of mitochondrial fragments is facilitated by DRP1 and Fis1 proteins. Mitophagy is mediated Parkin and PINK1 proteins. Cytochrome C is released from the damaged mitochondria in case of insufficient mitophagy, which induces apoptosis.

It has been observed that in heart failure the mitochondrial network of cardiomyocytes is disrupted and dynamics are shifted towards fission. Because of membrane instability during fission, the membrane potential between the two sides of the inner membrane decreases. This is associated with a deterioration in mitochondrial function, resulting in reduced amount of high-energy phosphates, decreased creatine phosphate/ATP ratio and impaired electron transport chain function, as well as increased production of oxygen free radicals and calcium levels [4, 13–16].

Initially, attempts were made to prevent the fission-induced decrease of membrane potential and deterioration of mitochondrial function, by inhibiting the fission processes themselves. The first and foremost investigated compound is Mdivi-1 or mitochondrial division inhibitor 1, that is a small molecule that has been studied for its potential role in inhibiting mitochondrial fragmentation. Mdivi-1 is thought to act by targeting a protein called dynamin-related protein 1 (DRP1), which plays a key role in mitochondrial fission [17]. The DRP1 protein inhibitor Mdivi-1 has protective effect against overload-induced acute heart failure and preventing a fission-induced decrease in membrane potential [17–19]. Moreover, it also has protective effect against doxorubicin-induced cardiac injury and in acute myocardial infarction and ischemia-reperfusion injury models [20–22].

It is important to note that despite the above-mentioned positive effects, there are no data regarding the long-term effect of Mdivi-1, moreover its safety and efficacy in humans are still areas of ongoing research. Recent data suggest that Mdivi-1 in cardiomyocytes can also cleave L-OPA1, decrease the expression of OXPHOS complex proteins, and elevate the superoxide production. These effects lead to impaired mitochondrial respiration and inhibition of macro-autophagy [23]. The inhibition of fission, and consequently the mitophagy, may not necessarily yield long-term benefits.

Therefore, in chronic stress scenarios instead of fission-inhibition, fusion promotion and stabilization of membrane structure came to the forefront of the research. It is assumed that shifting mitochondrial dynamics towards fusion may also have a positive effect on the outcome of heart failure. This hypothesis is supported by the fact that knocking out the fusion proteins OPA1 and Mfn2 caused heart failure [7, 24–27].

One of the proteins responsible for mitochondrial fusion is the long form of the GTPase family OPA1 protein (L-OPA1), which binds to the inner membrane and promotes fusion by soluble N-ethylmaleimide-sensitive factor attachment protein receptor (SNARE) complex formation after GTP hydrolysis, preventing membrane integrity and inhibiting pore formation. L-OPA1 is cleaved by proteases YME1L1 and OMA1, resulting in the formation of a soluble short form of OPA1 (S-OPA), which limits fusion and plays a role in mitophagy [28–30].

Several genetic animal models have been developed to study the effects of OPA1 protein. A muscle-specific OPA1 KO mutation in neonatal mice led to death within 9 days postnatally, so an inducible OPA1 knock-out model was also established. These animals showed signs of premature aging and died within 3 months of induction [31]. Using X chromosome-targeted Opa1 transgenesis, moderate OPA1 overexpression was observed in all cells [32]. These mice were fertile and had normal lifespan, with improved respiratory chain efficiency and they showed protective effects in certain genetic mitochondrial disease models [33, 34].

In this study we aimed to investigate and provide a basic cardiological characterization of the promotion of OPA-1 protein in mice.

In our animal model, we used heterozygous knock-in mice in which a ΔS1 mutant version of the Opa1 gene coding sequence was transfected. Therefore, in addition to the wild-type OPA1, mice also express a ΔS1 mutant version of OPA1 protein which cannot be cleaved by OMA1 protease. The OMA1 protease is activated by stress, mitochondrial damage and

membrane potential reduction and this protein is responsible for cleaving OPA1 at the S1 cleavage site. YME1L1 constitutively regulates OPA1 processing by cleavage at the S2 cleavage site. We suggest that deleting the OMA1 S1 cleavage site prevents the stress response-induced S-OPA1 dominance and consequent mitochondrial fragmentation. The stable L-OPA1 contributes to cristae structure stability, while the constitutively active YME1L1 maintains the necessary S-OPA1 levels for normal function, including normal fission induction and chaperone functions.

## Materials and methods

### Ethics statement

Animals received care according to the Guide for the Care and Use of Laboratory Animals published by the US National Institute of Health and the experiment was approved by the Animal Research Review Committee of the University of Pecs, Medical School (Permit number: BA02/2000-48/2022).

### Animal model

Mice were housed in controlled environments maintained at 20–24°C with 30–70% humidity and a 12-hour light/dark cycle in the Animal Research Facility of Szentagothai Research Center, University of Pecs. They were kept in spacious, durable cages providing at least 75 $cm^2$ per mouse, with soft bedding. Social housing is prioritized, with compatible groups to promote natural behaviors, and environmental enrichment is provided through nesting materials, tunnels, and toys.

To alleviate suffering during procedures (echocardiography) and sacrificing, mice are first placed in an induction chamber with isoflurane gas to ensure a gentle induction of anesthesia. During harvesting, a combination of ketamine (100 mg/kg) and xylazine (10 mg/kg) is administered for anesthesia, with maintenance using isoflurane. Once fully anesthetized, they undergo a thoracotomy, where the chest cavity is carefully opened. The heart is then removed and biological sampling is performed, ensuring the process is swift and humane to minimize distress and pain.

In our study OPA1 male transgenic mice (n = 10) were used, bred at the Department of Biochemistry and Medical Chemistry, University of Pecs, Medical School (permit number: KA-3440). In our model, we used knock-in mice that had the ΔS1 mutant version of the Opa1 gene sequence inserted into their DNA using a PiggyBac transposon gene expression vector. This mutant variant lacks the sequence encoding the S1 cleavage site (190–200 amino acids), resulting in mice expressing the ΔS1 mutant version of the OPA1 protein in addition to the wild-type OPA1 variant, which cannot be cleaved by OMA1 protease. The mutation is maintained in heterozygous. Wild-type littermates from transgenic animals were used as controls. Genotyping was performed at separation.

11-week-old OPA1 male transgenic mice (n = 10) and their wild-type littermates (n = 12) were used. Between 11 and 36 weeks of age echocardiography, blood pressure measurement and ECG were regularly performed. At the end of the experimental period, the animals were sacrificed, then hearts and lungs were removed. Atria and great vessels were trimmed from the ventricles, and the weight of the ventricles was measured. Hearts were fixed in 6% formalin for histology or freeze-clamped for Western blot analysis. Lungs were frozen to determine the wet/dry lung ratio.

## Echocardiographic measurements

Transthoracic echocardiography was performed under inhalation anesthesia at the beginning of the experiment (11 week) and on the day of sacrifice (36 week). The mice were lightly anesthetized with a mixture of 1.5% isoflurane and 98.5% oxygen. The chest of the animals was shaved, and acoustic coupling gel was applied. The animals were imaged in the left lateral position, and a warming pad was used to maintain normothermia. Heart rate did not differ considerably during anesthesia between the groups. Ventricular dimensions, wall thicknesses, and systolic functions were measured from parasternal short and long-axis views just above the midpapillary level. Parameters (E, A, and E') required for the evaluation of diastolic function were measured from the apical 4 chamber view. For the imaging of mice, VEVO 3100 high-resolution ultrasound imaging system (VisualSonics, Toronto, Canada) was used, which was equipped with a 25 MHz transducer. LV inner dimensions (LVIDd and LVIDs), LV end-diastolic volume (LVEDV), LV end-systolic volume (LVESV), E/A, and E/E' ratio were determined. EF (percentage) was calculated by $100 \times [(LVEDV - LVESV)/LVEDV]$.

## Non-invasive blood pressure measurement

Non-invasive blood pressure measurements were carried out on each animal at the beginning (11 week of age) and the at end of the study (36 weeks of age). Blood pressure measurement was performed by a non-invasive tail-cuff method using Hatteras SC1000 Blood Pressure Analysis System (Panlab, Harvard Apparatus).

## Determination of plasma B-type natriuretic peptide

After the sacrifice, blood samples were collected into Lavender Vacutainer tubes containing EDTA and aprotinin (0.6 IU/mL of blood) and were centrifuged at 1600g for 15 min at 4°C to separate the plasma. Supernatants were collected and BNP were determined by ELISA method as the manufacturer proposed (Mouse BNP ELISA Kit, Novus Biologicals, #NBP2-70011).

## Histology

Hearts were removed for histological examination at the end of the study. Ventricles were fixed in 6% formalin, sliced, and embedded in paraffin. Five-micrometer-thick sections were cut serially from the base to the apex by microtome. Six-six animals from both transgenic and wild-type groups and 3 sections from each heart were used to determine interstitial collagen deposition in the heart. Images (magnification 10x) were randomly taken from the middle region of the left ventricle wall on each section. The fibrotic area was determined on each image, and the mean value of nine images represents each animal. Masson's Trichrome staining was applied to visualize interstitial fibrosis. Fifteen-month-old mice were used as positive controls. Picrosirius red dye was used to measure cardiomyocyte diameter as a cellular marker of myocardial hypertrophy. Six animals from both groups and three sections from each animal were used to determine the cell diameter. Images (magnification 10x) were randomly taken from the free LV wall on each section. The fitted polygon technique was used to determine the area of the cells and the calculated diameter was used for statistical analysis. 100 cardiomyocytes were measured from each animal in order to evaluate the cardiomyocyte diameter and each group contained 6 animals.

 Immunohistochemical staining was used to determine the extent of oxidative damage. Monoclonal anti-nitrotyrosine antibody (Abcam, ab125106) was used to measure oxidative damage of the proteins and anti-8-oxoguanine antibody (Abcam, ab206461) was used to measure oxidative DNA damage. Binding was visualized with a biotinylated secondary antibody

followed by the avidin–biotin-peroxidase detection system using DAB as chromogen. Progress of the immunoreaction was monitored under a light microscope and the reaction was stopped by the removal of DAB. Six-six animals from both transgenic and wild-type groups and 3 sections from the free wall of the left ventricle of each heart were used to determine the degree of oxidative damage to the heart. ImageJ software was used to determine the extent of DAB staining on 20x magnification images.

## Electron microscopic examination of hearts

For electron microscopic analysis, hearts were perfused retrogradely through the aortic root with ice-cold PBS to wash away blood and then washed by modified Kranowsky fixative (2% paraformaldehyde, 2.5% glutaraldehyde, 0.1 M Na-cacodylate buffer, pH 7.4 and 3 mM CaCl2). The electron microscopic samples were prepared as described earlier [35], and they were examined with a JEM 1200EX-II electron microscope. Four animals of each group, 3–5 blocks from each animal were used. The area of interfibrillar mitochondria (IFM) were measured by free hand polygon selection (n ~ 500/group) using the NIH ImageJ software.

## Neonatal Mouse Cardiomyocyte (NMCM) cell culture

Cardiomyocytes were isolated using the Pierce™ Primary Cardiomyocyte Isolation Kit (Life Technologies, Carlsbad, CA, USA #88281) from 1-3-day old neonatal OPA1 transgenic mice and their wild-type littermates. The isolation of cardiomyocytes was performed as described earlier [36].

## Western-blot analysis

**Western blot sample preparation from cardiac tissue.** Thirty milligrams of lateral ventricle wall from 8–8 animals were homogenized in ice-cold Tris buffer (50 mmol/l, pH 8.0) containing protease inhibitor (1:100; Sigma-Aldrich Co., #P8340) and phosphatase inhibitor (1:100; Sigma-Aldrich Co. #P5726). The supernatants were harvested in 2x concentrated sodium dodecyl sulphate- (SDS-) polyacrylamide gel electrophoresis sample buffer.

**Electrophoresis and transfer of proteins.** Proteins were separated on 10% SDS-polyacrylamide and transferred to nitrocellulose membranes. After blocking (2 h with 2% non-fat dry milk in Tris-buffered saline), membranes were probed overnight at 4°C with primary antibodies recognizing the following antigens: glyceraldehyde 3-phosphate dehydrogenase GAPDH; 1:1000; Cell Signaling #2118) as a loading control, optic atrophy 1 (OPA1(D6U6N); 1:1000; Cell Signaling #80471), DYKDDDD Tag (FLAG; 1:500; Invitrogen #PA1-984B), voltage-dependent anion channel (VDAC; 1:1000; Cell Signaling #4866), Bcl-2/E1B-19kDa interacting protein 3 (BNIP3; 1:1000; Cell Signaling #3769), mitofusin 1 (Mfn1(11E91H1); 1:200; Abcam #ab126575), mitofusin 2 (Mfn2(D2D10); 1:1000; Cell Signaling #9482), dynamin-related protein 1 (DRP1(4E11B11); 1:1000; Cell Signaling #14647), parkin (PRK8; 1:2000; Abcam #77924), PINK1 (N4/15; 1:1000; Abcam #186303), OMA1 (1:2000; Proteintech #17116-1-AP), YME1L 1(1:1000; Proteintech #11510-AP).

## Evaluation of mitochondrial fragmentation with fluorescent microscopy

NMCM cells were seeded at a density of $2x10^4$ cells/well in 24 well glass bottom plates and cultured at least 3 days before the experiment. On the day of the experiment, cells were washed once in HBSS and were added 100 nM MitoTracker Red CMXRos dissolved in serum-free DMEM and incubated for 30 min at 37°C. After the incubation, the cells were washed with HBSS, and the mitochondrial network was visualized by Nikon Eclipse Ti-U fluorescent

microscope equipped with a Spot RT3 camera using a 60x objective and epifluorescent illumination. We conducted a thorough quantification analysis of the mitochondrial shapes using the Mitochondria Network Analysis tool (MiNA) on the ImageJ interface. For this analysis, images were imported into ImageJ and processed into 8-bit grayscale images. To enhance the quality, we applied an unsharp mask, enhanced local contrast, and performed median filtering. The images were then binarized to generate black foreground mitochondria images against a white background and converted to a skeleton that represents the features of the original mitochondria image in the form of lines. The skeletons were further analyzed using the ImageJ "analyze skeleton" plugin, which measures the length of each branch.

## Mitochondrial membrane potential measurement with JC-1 assay

The mitochondrial membrane potential ($\Delta\Psi$m) was measured using the mitochondrial membrane potential specific fluorescent probe, JC-1 (Enzo Life Sciences, ENZ-52306) as we performed earlier [48]. When excited at 488 nm, the dye emits red fluorescence (590 nm) at high $\Delta\Psi$m and green (530 nm) at low $\Delta\Psi$m. NMCM cells were seeded on glass bottom 24 well plate and seeded at least 3 days before the experiment. The cells were washed in HBSS and incubated for 90 minutes at 37°C in media containing 5 μg/mL JC-1. We used a pharmacological control group in which wild-type NMCM cells were treated with 10 μM FCCP for 90 minutes. Following incubation, the cells were washed once with HBSS and then imaged with Nikon Eclipse Ti-U fluorescent microscope equipped with a Spot RT3 camera using a 60x objective and epifluorescent illumination. We examined only beating cells, all measurements were repeated three times. Fluorescent signals were quantified by using the ImageJ software (NIH, Bethesda, MD, USA).

## Evaluation of the mitochondrial energy metabolism and function

As we described earlier, Agilent Seahorse Extracellular Flux (XFp) Analyser (Agilent Technologies, (Santa Clara, CA, USA)) was used to determine the NMCM cells' oxygen consumption rate (OCR) [36]. NMCM cells were seeded in XFp Miniplate at a density of $4 \times 10^4$ cells/well in 80 μL complete growth medium (DMEM for Primary Cell Isolation containing 10% FBS, 100 IU/mL penicillin and 100 μg/mL streptomycin) and incubated at 37°C, 5% $CO_2$ for 2 days. On the day before the experiment, sensor cartridges were hydrated in XFp calibrant and maintained at 37°C without $CO_2$ overnight. On the day of the assay, DMEM for Primary Cell Isolation medium was replaced by Agilent Seahorse XF Base Medium containing 1 mM pyruvate, 2 mM glutamine and 10 mM glucose (adjusted pH to 7.4 with 0.1 N NaOH). Before measurement, different compounds were loaded into the appropriate ports of a hydrated sensor cartridge (10 μM oligomycin, 10 μM FCCP and 5 μM rotenone/antimycin). Three measurements were performed after each injection. OCR was used to determine mitochondrial energy metabolism. The parameter values, including basal respiration, maximal respiration, ATP-associated OCR and spare respiratory capacity, were determined according to the Seahorse XFp Cell Mito Stress user guide protocol. Data were analyzed using the Seahorse XF test report analysis.

## mtDNA copy number

Total DNA was isolated from lateral ventricle wall from 8–8 animals using GenElute™ Mammalian DNA Miniprep kit (Sigma Aldrich Co., St. Louis, MO, USA). For quantification of mtDNA copy number, real-time PCR analysis was performed with the NovaQUANT™ Mouse Mitochondrial to Nuclear DNA Ratio Kit (Sigma-Aldrich Co., St. Louis, MO, USA) according to the manufacturer's instructions. A set of four optimized PCR primer pairs targeting two mitochondrial genes (trLEV and 12s RNA) and two nuclear genes (BECN1 and NEB) were

pre-aliquoted in an Applied Biosystems MicroAmp® Fast Optical 96-well Reaction Plate. Real-time DNA amplification was performed using a Bio Rad CFX96 Touch Real-Time PCR Detection System. The results of the qPCR reactions were analyzed with 2 −ΔCT method.

## Statistical analysis

Statistical analysis was performed by SPSS for Windows, version 26.0. All data were expressed as a mean ± standard error of mean (SEM) of the replicate measurements. The normality of distribution was assessed by the Shapiro-Wilk test. The homogeneity of the groups was tested by Levene's test. A Student's t-test was used to compare the mean values from the two groups. A value of $p < 0.05$ was considered statistically significant.

## Results

### Effect of OPA1 transgenic phenotype on gravimetric parameters, systolic blood pressure, BNP level and echocardiographic values during aging

The gravimetric parameters did not differ significantly between the groups, neither at the beginning nor at the end of the study. There was no significant difference in the systolic blood pressure values between the transgenic mice and their wild-type littermates, either at 11 weeks or at 36 weeks. There was no convincing difference in the BNP level, determined from serum, at the end of the experimental period (Table 1).

At the beginning of the study, the left ventricular end-diastolic volume was significantly higher in transgenic animals compared to wild-type mice ($p < 0.05$ TG$^{START}$ vs. WT$^{START}$), this difference disappeared by the end of experiment (Table 2). At the age of 36 weeks, we observed a significant decrease in the ejection fraction in the transgenic animals ($p < 0.01$ TG$^{END}$ vs. TG$^{START}$), while the cardiac function of the wild-type animals remained normal. The left ventricular end-systolic volume (LVSV) increased in both groups, this increase was more pronounced in the transgenic group, but there was no statistically significant difference between the wild-type and transgenic groups. The diastolic function marker E/E' ratio increased in transgenic animals indicating a worsening diastolic function, but this change did not reach the level of significance. Significant parameter changes that occur naturally with growth are not marked.

**Table 1. Effect of OPA1 transgenic phenotype on gravimetric parameters, systolic blood pressure and BNP level during aging.** BW$^{START}$: body weight at age of 11-week, BW$^{END}$: body weight at age of 36-week, VW$^{END}$: ventricles weight at the end of the study, TL: length of the right tibia, SBP$^{START}$: systolic blood pressure at age of 11-week, SBP$^{END}$: systolic blood pressure at age of 36-week, BNP: B-type natriuretic peptide, WT: wild-type mice (n = 12), TG: OPA1 transgenic mice (n = 10). Values are means ± SEM.

| | WT (n = 12) | TG (n = 10) |
|---|---|---|
| **BW$^{START}$** (g) | 28.67 ± 1.03 | 27.32 ± 1.81 |
| **BW$^{END}$** (g) | 32.15 ± 1.24 | 31.62 ± 2.48 |
| **VW$^{END}$** (mg) | 142.03 ± 3.25 | 143.32 ± 6.97 |
| **VW/BW$^{END}$** (mg/g) | 4.45 ± 0.13 | 4.73 ± 0.36 |
| **VW/TL** (mg/mm) | 6.84 ± 0.19 | 7.07 ± 0.32 |
| **Lung wet weight/dry weight** | 5.07 ± 0.12 | 4.74 ± 0.24 |
| **SBP$^{START}$** (Hgmm) | 109.35 ± 2.14 | 110.47 ± 2.41 |
| **SBP$^{END}$** (Hgmm) | 110.67 ± 2.41 | 112.67 ± 1.74 |
| **BNP** (pg/ml) | 106.41 ± 7.25 | 95.11 ± 11.25 |

**Table 2. Effect of OPA1 transgenic phenotype on echocardiographic parameters during aging.** LVIDd: left ventricular (LV) inner diameter end-diastolic; LVIDs: LV inner diameter end-systolic; LVEDV: LV end-diastolic volume; LVES: LV end-systolic volume; LV mass: calculated weight of left ventricle; EF: ejection fraction; E: mitral peak velocity of early filling; A: mitral peak velocity of late filling; E': early diastolic mitral annular velocity; WT$^{START}$: wild-type littermates at 11-week of age, WT$^{END}$: wild-type littermates at 36-week of age; TG$^{START}$: OPA1 transgenic mice at 11-week of age, TG$^{END}$: transgenic mice at 36-week of age. *p<0.05 vs. WT$^{START}$, **p<0.01 vs. TG$^{START}$. Values are means ± SEM.

| | WT$^{START}$ (n = 12) | WT$^{END}$ (n = 12) | TG$^{START}$ (n = 10) | TG$^{END}$ (n = 10) |
|---|---|---|---|---|
| **LVIDd** (mm) | 4.07 ± 0.12 | 4.29 ± 0.11 | 4.35 ± 0.17 | 4.29 ± 0.17 |
| **LVIDs** (mm) | 2.99 ± 0.14 | 3.19 ± 0.18 | 3.02 ± 0.15 | 3.29 ± 0.19 |
| **LVEDV** (μl) | 65.82 ± 4.57 | 86.22 ± 5.17 | 89.03 ± 8.02* | 90.58 ± 8.27 |
| **LVESV** (μl) | 30.95 ± 3.02 | 40.74 ± 4.38 | 37.49 ± 4.27 | 49.26 ± 6.86 |
| **LV MASS** (mg) | 110.97 ± 4.16 | 156.56 ± 10.39 | 136.94 ± 13.33 | 159.94 ± 15.62 |
| **EF** (%) | 57.23 ± 3.35 | 51.94 ± 3.04 | 58.6 ± 2.59 | 44.69 ± 3.56** |
| **E/A** | 1.75 ± 0.08 | 2.48 ± 0.45 | 2.02 ± 0.13 | 2.3 ± 0.16 |
| **E/E'** | 30.21 ± 2.08 | 27.63 ± 2.65 | 28.69 ± 4.92 | 36.72 ± 4.93 |

## Effect of OPA1 transgenic phenotype on interstitial collagen deposition

Picrosirius red staining was performed on the left ventricles in order to monitor the degree of fibrosis (Fig 2). Only a low amount of interstitial collagen deposition could be seen in both transgenic and wild-type mice and there was no significant difference between the groups (WT: 12.99 ± 0.70% vs. TG: 11.92 ± 0.98%). Positive control group has significant higher level of collagen deposition (Positive control: 23,7±0.53%).

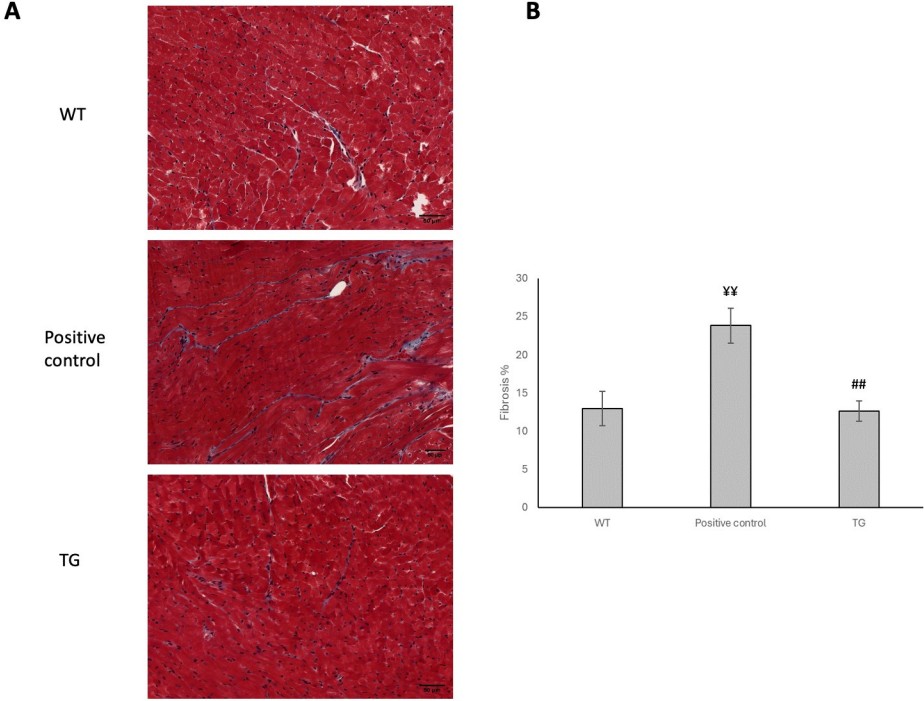

**Fig 2. Effect of OPA1 transgenic phenotype on the extent of interstitial fibrosis at the age of 36-weeks.** Representative histological sections stained with Masson's trichrome (A) (n = 6). Scale bar: 50 μm, magnification: 10-fold. Densitometric evaluation of the section is shown (B). Data are expressed as mean ± SEM. WT vs Positive Control ¥¥ p < 0.01, TG vs Positive Control ## p < 0.01.

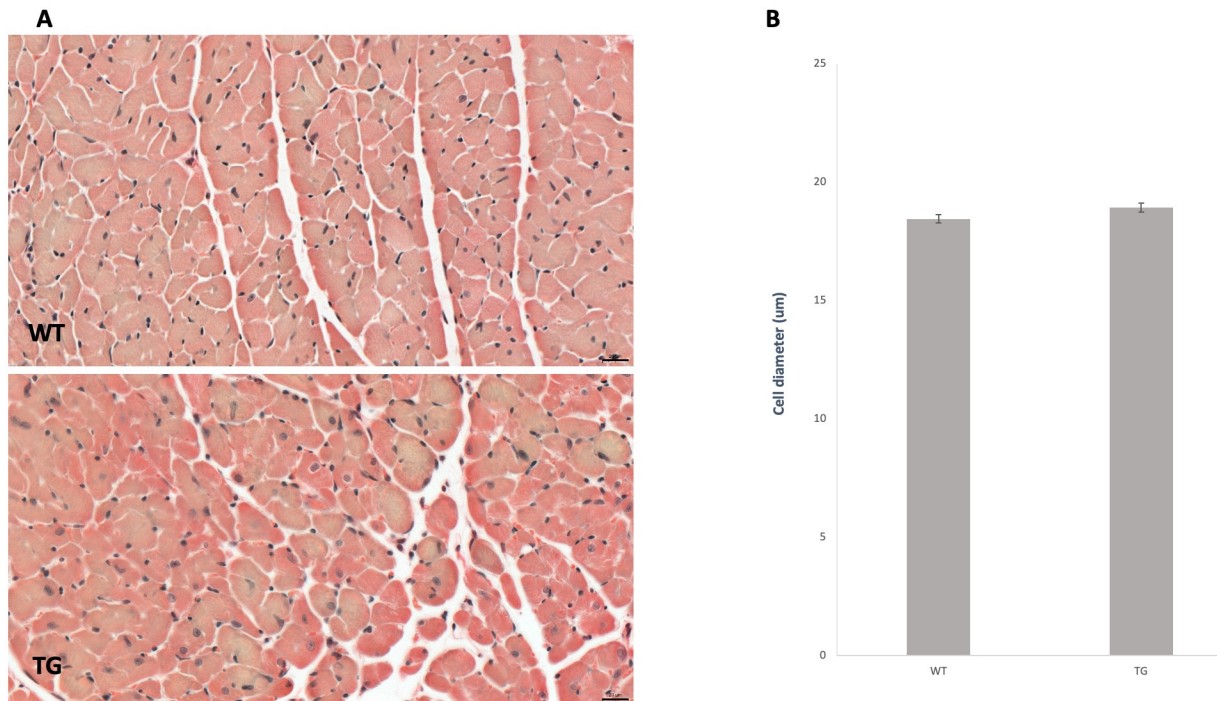

**Fig 3. Effect of OPA1 transgenic phenotype on cardiomyocyte cell diameter at the age of 36 weeks.** Representative histological sections stained with Picrosirius red (A) (n = 6). Scale bar: 20 μm, magnification: 40-fold. The average cellular diameter in the two groups is shown (B). WT: wild-type mice, TG: OPA1 transgenic mice.

### Effect of OPA1 transgenic phenotype on the diameter of cardiomyocytes

Histological samples from the left ventricle of the heart stained with Picrosirius red were used to study the cell diameters (Fig 3). The diameter of cardiomyocytes did not differ between groups (WT: 18.45 ± 0.18 vs. WT: 18.93 ± 0.5).

### Effect of OPA1 transgenic phenotype on oxidative damage of proteins and DNA

For verification of oxidative stress, the production of nitrotyrosine and 8-Oxoguanine was investigated by immunohistochemical staining (Fig 4). No difference was found between the wild-type and the OPA1 transgenic group either by anti-nitrotyrosine staining specific for oxidative damage to proteins (WT: 17.8 ± 2.7%, TG: 16.4 ± 1.4%) or by anti-8-oxoguanine staining specific for oxidative damage to DNA (WT: 6.7 ± 0.4%, TG: 6.5 ± 0.5%).

### Effect of OPA1 transgenic phenotype on mitochondrial ultrastructure

Longitudinal sections of myofibrils were evaluated to assess the status of the interfibrillar mitochondria by electron microscopy (Fig 5). The area of interfibrillar mitochondria was examined on electron micrographs (500 mitochondria/group were measured). We determined the relative frequencies of the measured mitochondrial areas in arbitrary intervals of 0.3 μm$^2$. In both group, the predominant area range of the measured mitochondria was between 0.3 and 0.6 μm$^2$ (WT: 48.4% vs. TG: 42.8%). However, in the transgenic group, the size of the mitochondria was more heterogeneous. We found a higher proportion of mitochondria in both <0.3 μm$^2$ (WT: 10.4% vs. TG: 13.6%) and the >0.9 μm$^2$ ranges (WT: 13.2% vs. TG: 19.6%)

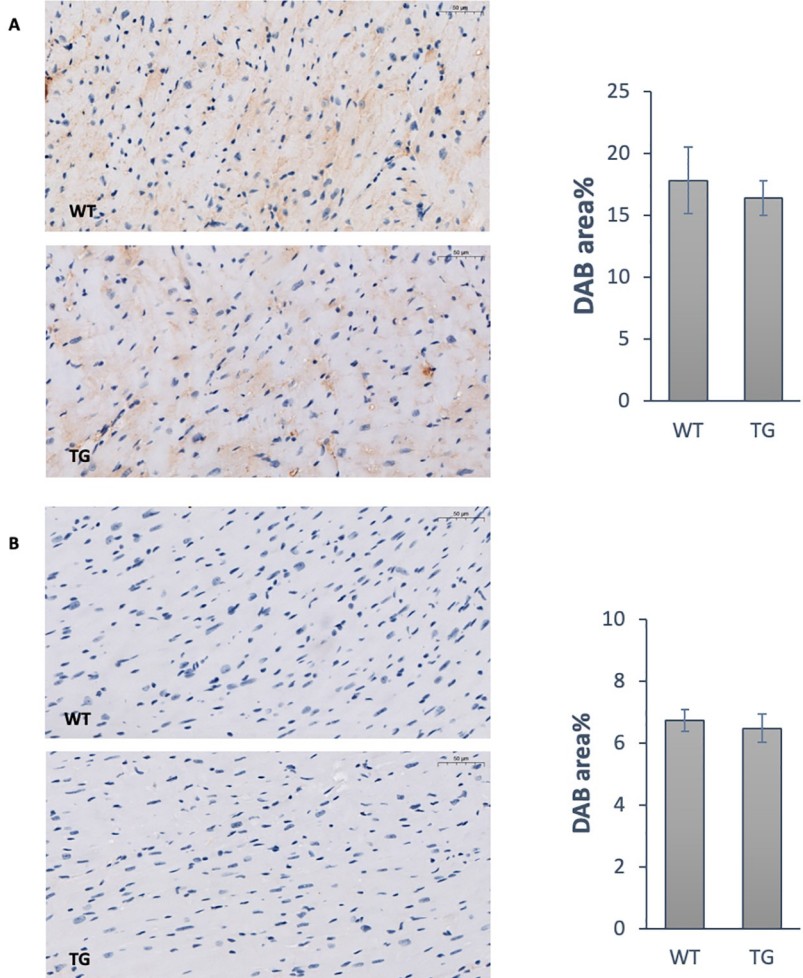

**Fig 4. Effect of OPA1 transgenic phenotype on the oxidative stress.** Representative immunohistochemical staining and densitometric evaluation of the section for nitrotyrosine (A) and 8-Oxoguanine (B) in the heart (n = 6). Scale bar: 50 μm, magnification 20-fold. WT: wild-type mice, TG: OPA1 transgenic mice.

than in the wild-type (Fig 5F). The mean mitochondrial area in transgenic animals was significantly larger than in wild-type (WT: 0.606 μm$^2$ vs. TG: 0.684 μm$^2$, p<0.05) (Fig 5E). Smaller vacuole-like intercristal widenings and a minor disruption of mitochondrial cristae structure could be observed in the transgenic group, moreover the mitochondrial matrix was also lighter (Fig 5D).

## Morphological analysis of mitochondrial network

We conducted a thorough quantification analysis of the mitochondrial shapes using the Mitochondria Network Analysis tool (MiNA) on the ImageJ interface. The ultrastructurally observed larger mitochondrial area was not reflected in the characteristics of the mitochondrial network; the branch length of the mitochondrial network did not differ between the wild-type and transgenic animals (Figs 6 and 7).

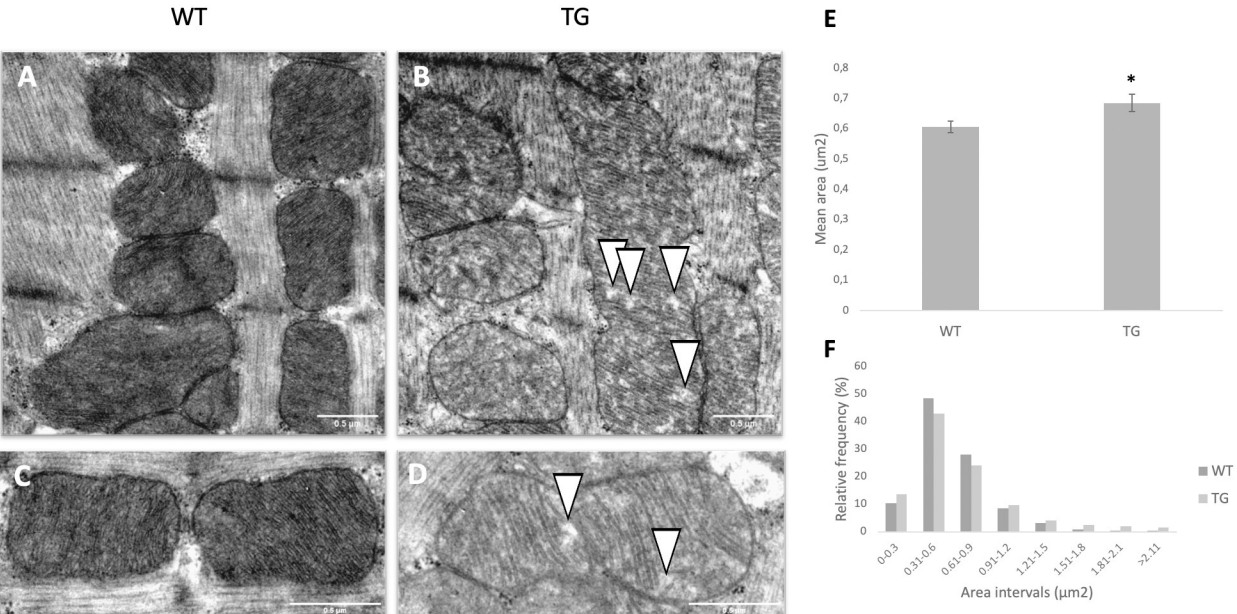

**Fig 5. Effect of OPA1 transgenic phenotype on interfibrillar mitochondria of the myocardium.** (A-B) Representative electron micrograph of interfibrillar mitochondria of wild-type (A) and OPA1 transgenic (B) mice (magnification 12 k, scale bar: 0.5 μm). (C-D) Ultrastructure of interfibrillar mitochondria in the myocardium of wild-type (C) and transgenic (D) animals (magnification: 25 k, scale bar: 0.5 μm). (E) Means of area values in given groups (~500 mitochondria/group). (F) Relative frequencies of measured mitochondrial areas in each arbitrary interval. WT: wild-type mice, TG: OPA1 transgenic mice. Data are expressed as mean ± SEM. * $p < 0.05$ vs. WT, n = 6.

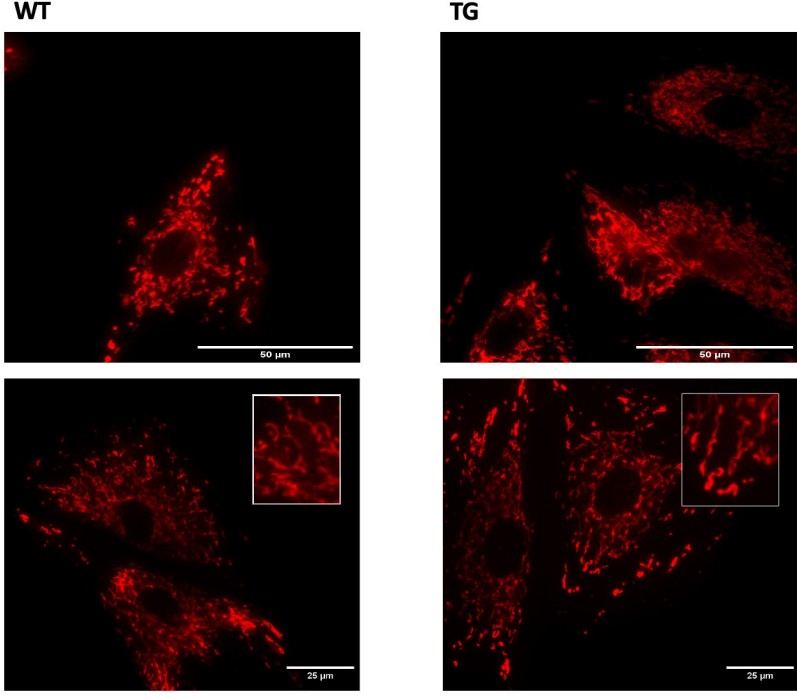

**Fig 6. Fluorescent staining of the mitochondrial network.** The neonatal mice cardiomyocyte cells were stained with 100 nM MitoTracker Red CMXRos, scale bar: 25 μm, magnification: 60-fold. Groups: WT: wild-type cardiomyocytes, TG: NMCM cells from OPA1 transgenic mice.

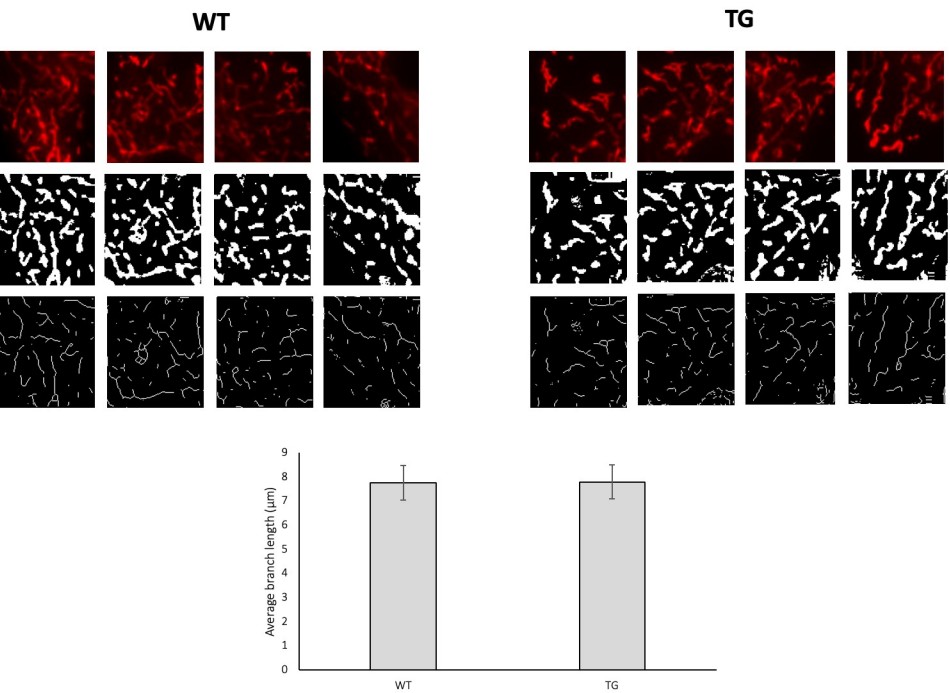

**Fig 7. Quantification analysis of the mitochondrial branch length.** The neonatal mice cardiomyocyte cells were stained with 100 nM MitoTracker Red CMXRos, Mitochondria Network Analysis tool (MiNA) on the ImageJ interface. scale bar: 25 μm, magnification: 60-fold. Groups: WT: wild-type cardiomyocytes, TG: NMCM cells from OPA1 transgenic mice.

## Effect of OPA1 transgenic phenotype on membrane potential (ΔΨ) in NMCM cells

We examined the mitochondrial membrane potential using JC-1, a cell-permeable voltage-sensitive fluorescent mitochondrial dye (Fig 8). JC-1 emits red fluorescence if the mitochondrial membrane potential is high (aggregated dye), while depolarized mitochondria emit green fluorescence (monomer dye). We observed that the mitochondrial membrane potential was significantly different between the groups. Stronger green and weaker red fluorescence was detected in the transgenic NMCMs compared to the wild-type NMCMs ($p<0.01$). As the validation of the method, FCCP treatment resulted in significantly stronger green and weaker red fluorescence in wild-type NMCMs.

## Effect of OPA1 transgenic phenotype on mitochondrial oxygen consumption and energy metabolism in NMCM cells

Neonatal cardiomyocyte cells were isolated from transgenic animals and from their wild-type littermates. To determine the mitochondrial energy metabolism and respiratory function, we used the Agilent Seahorse XFp Analyser system and Agilent Seahorse XFp Cell Mito Stress test (Fig 9). We observed that the oxygen consumption rate of the cells from the transgenic animals was lower compared to the wild-type cells. The deterioration was significant for maximal respiration (WT: 204.4 ± 15.05 pmol/min vs. TG: 152.21 ± 16.22 pmol/min, $p<0.05$) and spare respiratory capacity (WT: 142.7 ± 11.29 pmol/min vs. TG: 99.83 ± 10.89 pmol/min, $p<0.05$).

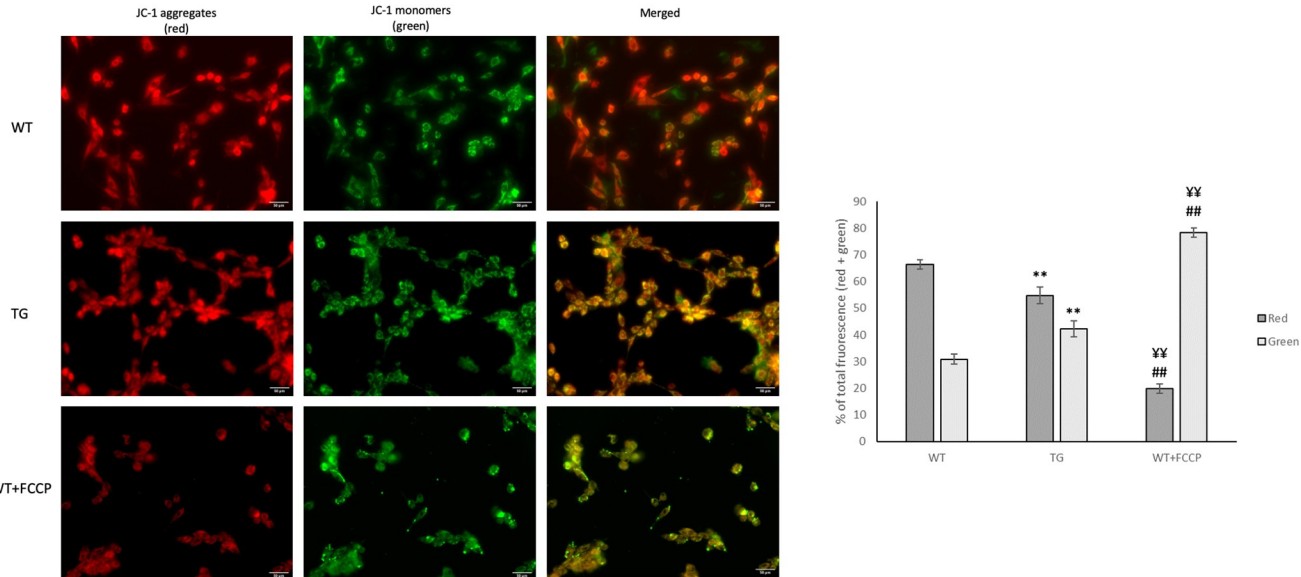

**Fig 8. Effect of OPA1 transgenic phenotype on mitochondrial membrane potential in NMCM cells.** Cells were stained with 5 μg/mL of JC-1, which is a membrane potential sensitive dye. The dye was loaded and after 90 minutes incubation, fluorescent microscopic images were taken using both the red and green channels. A: representative images are presented. Scale bar: 50 μm, magnification: 40-fold, WT: wild-type cardiomyocytes, TG: NMCM cells from OPA1 transgenic mice. WT+FCCP: wild-type NMCM cells treated with 10 μM FCCP. B: quantitative analysis of mitochondrial polarization. Data are presented as the mean ± SEM of four independent measurements, n = 6. WT vs TG **p<0.01, WT vs WT+FCCP ##p<0.01, TG vs WT+FCCP ¥¥ p<0.01.

## Effect of OPA1 transgenic phenotype on mtDNA copy number

To more precisely support the effect of OPA1 promotion on mitochondrial copy number, real-time PCR studies were performed. We found that the average number of mitochondrial

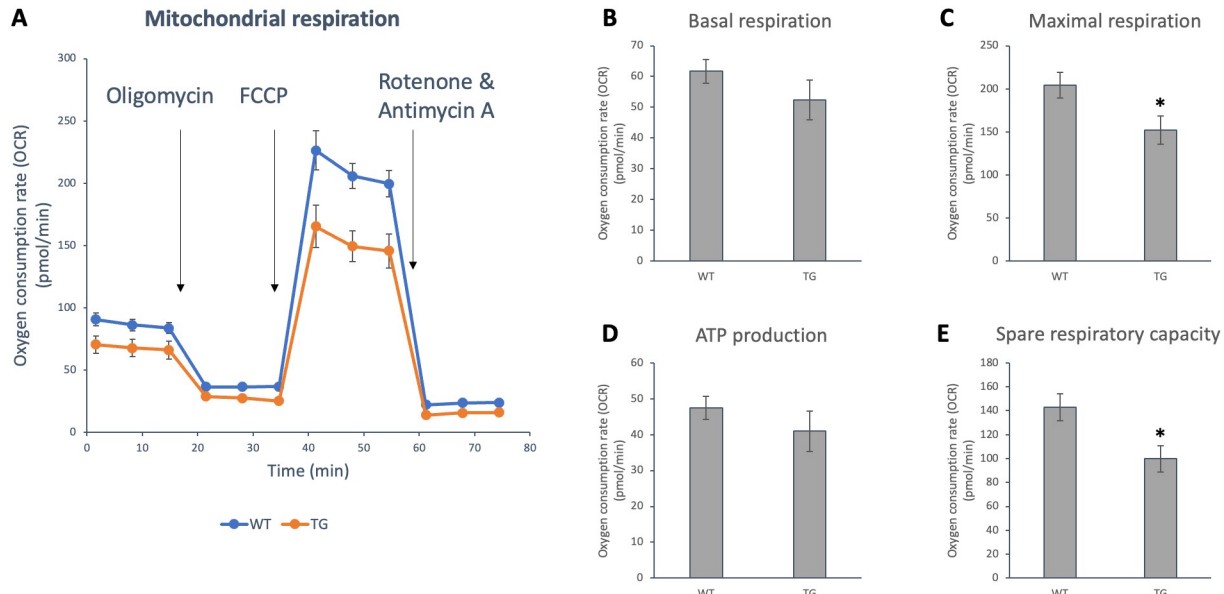

**Fig 9. Changes in the oxygen consumption rate of OPA1 transgenic NMCM cells.** A: oxygen consumption rate (OCR) in NMCM cells, measured by Seahorse XFp Analyser, B: basal respiration, C: maximal respiration, D: ATP production and E: spare respiratory capacity, WT: wild-type cardiomyocytes, TG: NMCM cells from OPA1 transgenic mice. *p<0.05 vs. WT. Values are means ± SEM, n = 6.

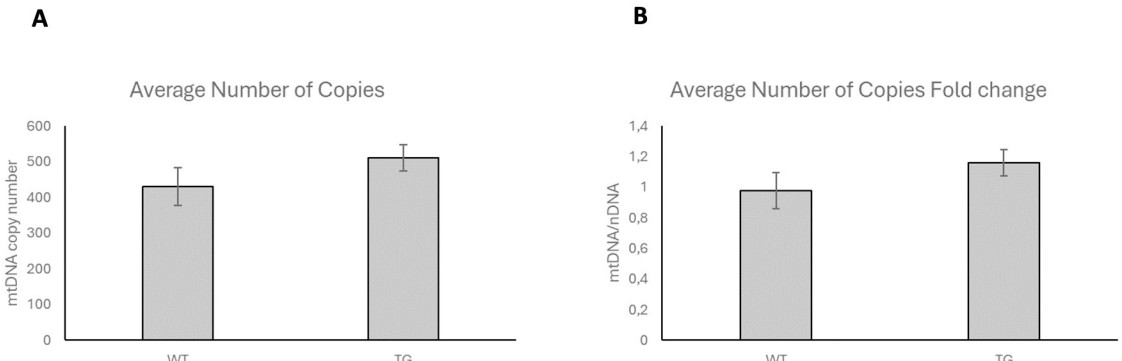

**Fig 10. Effect of OPA1 transgenic phenotype on mtDNA copy number.** A: Average number of copies, B: Average number of copies fold change, Using GenElute™ Mammalian DNA Miniprep kit For quantification of mtDNA copy number, real-time PCR analysis was performed with the NovaQUANT™ Mouse Mitochondrial to Nuclear DNA Ratio Kit. WT: wild-type, TG: OPA1 transgenic mice. Values are means ± SEM, n = 8.

copies was slightly increased in the TG group, but this increase did not reach statistical significance (Fig 10).

## Effect of OPA1 transgenic phenotype on the proteins of the mitochondrial dynamics

The OPA1 level of transgenic animals was significantly increased compared to the wild-types (p < 0.01 vs. WT group; Fig 11B). In the transgenic group primarily the amount of short

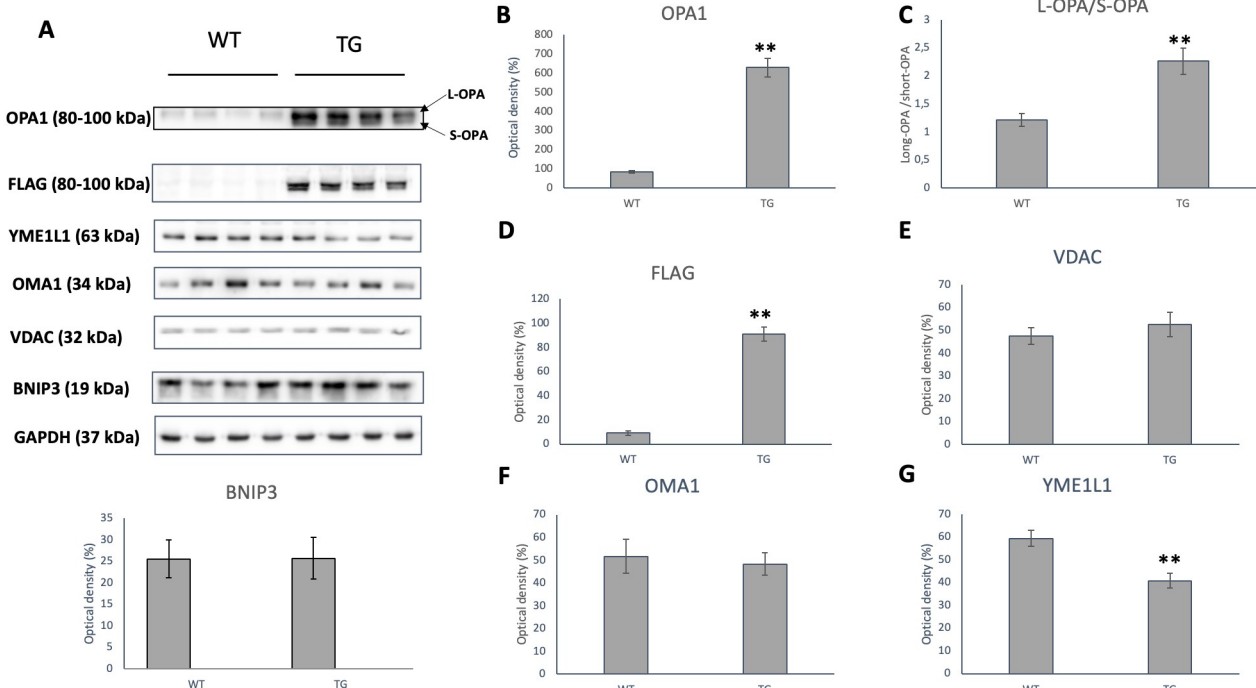

**Fig 11. Changes of OPA1, OMA1 and YME1L1 level Representative Western blot analysis of OPA1, DYKDDDDK Tag (FLAG), VDAC, BNIP3, OMA1, YME1L1 and densitometric evaluation are shown.** GAPDH was used as a loading control. WT: wild-type mice(n = 8), TG: OPA1 transgenic mice (n = 8). Values are mean ± SEM. ** p < 0.01 vs. WT.

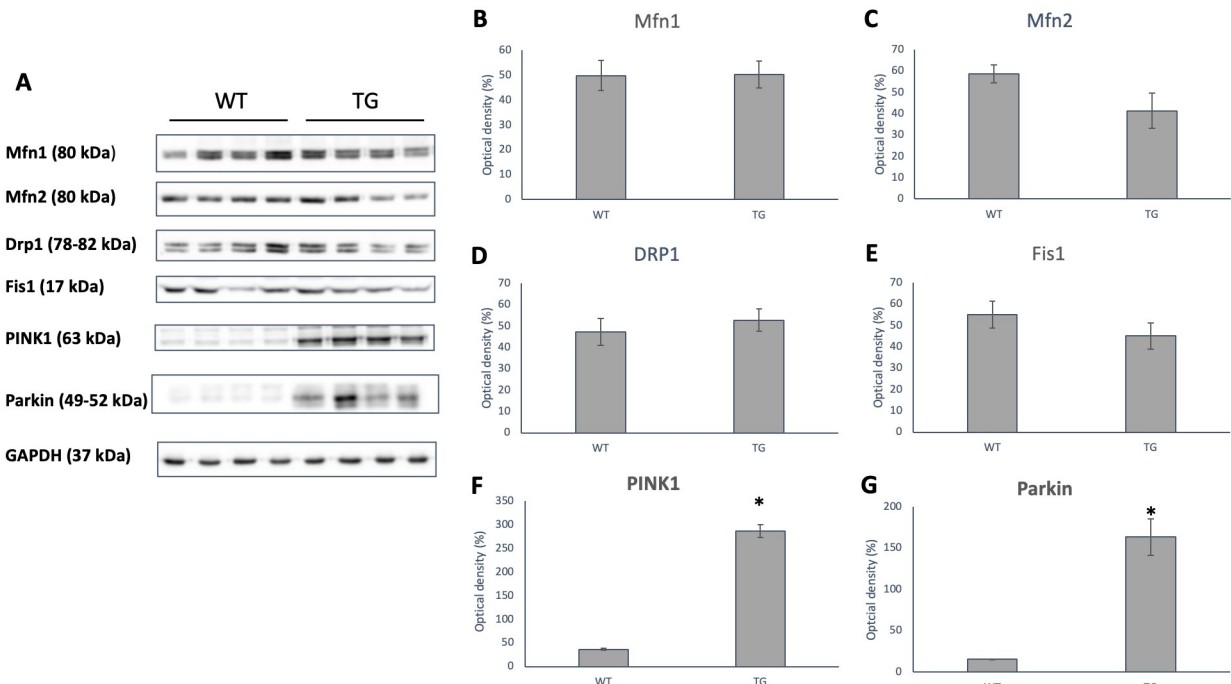

**Fig 12. Changes in protein levels of mitochondrial dynamics.** Representative Western blot analysis of Mfn1, Mfn2, DRP1, Fis1, PINK1, Parkin and densitometric evaluation are shown. GAPDH was used as a loading control. WT: wild-type mice (n = 8), TG: OPA1 transgenic mice (n = 8). Values are mean ± SEM. * p < 0.01 vs. WT.

OPA1 increased, the L-OPA1/S-OPA1 ratio was significantly lower (p < 0.01 vs. WT group; Fig 11C). Transgenic origin of the elevated OPA1 level was demonstrated with using DYKDDDDK Tag (FLAG) antibody (p < 0.01 vs. WT group; Fig 11D). The amount of VDAC, indicating total mitochondrial mass, did not differ significantly between the two groups (Fig 11E). Levels of proapoptotic mitochondrial BNIP3 did not differ between groups (Fig 11A Lower panel). The level of OMA1 protease was similar in the transgenic and wild-type animals (Fig 11F), while the expression level of YME1L1 was significantly lower in the transgenic animals (p < 0.01 vs. WT group; Fig 11G).

There was no significant difference between the groups in the case of Mfn1, but Mfn2 showed a decreasing tendency in the transgenic animals (Fig 12B and 12C). There was no difference in the level of fission proteins DRP1 and Fis1 between the two groups (Fig 12D and 12E). The amount of PINK1 and Parkin proteins, which play a role in mitophagy, increased in the transgenic group, in the case of PINK1 the change was significant (p < 0.05; Fig 12F and 12G).

## Discussion

It has been previously described that heart failure is accompanied by morphological and functional abnormalities of mitochondria and that mitochondrial alterations cause heart failure [4, 13]. The OPA1 level decreases in heart failure and conversely, low OPA1 level can manifest in impaired heart function [37–39]. A decrease in OPA1 level was also observed during aging, which was associated with a deterioration in muscle function. Early aging phenotype and early death were observed in an inducible muscle specific OPA1 KO mouse model [31]. Based on

the literature, we hypothesized that OPA1 overproduction has a beneficial effect on the changes that occur in aging by increasing the stability of the mitochondrial membrane [40].

OPA1 protein is expressed in eight different isoforms in humans and in four isoforms in mice through alternative splicing [41–43]. After translation, OPA1 proteins are transported into the mitochondria and anchor to the mitochondrial inner membrane. Two L-OPA1 proteins of two individual mitochondria can connect by homotypic protein interaction which promotes fusion and supports the cristae structure. The L-OPA1 can bind to a cardiolipin molecule of another mitochondrion and this heterotypic interaction also initiates GTP-hydrolysis-dependent fusion, but the presence of OPA1 is essential for this fusion type [44–46]. This L-OPA1 form undergoes further proteolytic cleavages, thereby creating soluble short forms [47, 48]. The OMA1 protease is activated by stress, mitochondrial damage and membrane potential reduction and this protein is responsible for cleaving OPA1 at the S1 cleavage site. The resulting short OPA1 has a multifaceted task. It inhibits fusion, thereby playing an important role in the elimination of damaged mitochondria [28, 29]. On the other hand, it acts also as a chaperone, protecting the proteins of the intermembrane space from thermal and chemical aggregation [49, 50]. OMA1 is inactivated by autocatalysis, when the triggering effect is terminated, making the process reversible [51]. OMA1 can also contribute to the integrated stress response of the cell by cleaving DAP3-binding cell death enhancer 1 (DELE1). This OMA1-- mediated protective response was shown to be essential for the survival of mice in a knock-in model of mitochondrial myopathy in which an abnormal protein accumulated the inner membrane [52]. In contrast, YME1L1 constitutively regulates OPA1 processing by cleavage at the S2 cleavage site. YME1L1 also plays an important role in cellular quality control, organelle biogenesis, membrane fission and vesicular transport [53–55]. The anchored L-OPA1 and the soluble S-OPA1 forms are in dynamic equilibrium, interacting with each other to regulate the fission and fusion processes [47, 51, 53, 55, 56].

It has been proposed that stress-triggered cleavage of OPA1 leads to fragmentation of mitochondria and enhances cell sensitivity to death. However, a recent research has revealed that S-OPA1 possesses the ability to preserve cristae structure and maintain energetic functions due to its GTPase activity, despite its lack of fusion capability [57]. These data indicates that the production of S-OPA1 via OPA1 cleavage may serve as a protective mechanism in cells under stress conditions [57].

In our model, we discovered that there are higher level of OPA-1, and the ratio of L-OPA/ S-OPA is skewed towards the long form. However, the level of S-OPA was higher in the transgenic group in comparison to wild-type animals.

The question arises: Could this phenomenon combine the beneficial effects of both variants?

We performed our experiments on transgenic knock-in mice, in which the ΔS1 mutant version of the coding sequence of the Opa1 gene was inserted. The OPA1 protein expressed in this way lacks S1 cleavage site, so this protein cannot be cleaved by the OMA1 protease. As a result, the transgenic mice express the ΔS1 mutant version of the OPA1 protein in addition to the wild-type OPA1 version. The mutation was maintained in a heterozygous form to moderate the excessively high OPA1 level, which would inhibit the removal of damaged mitochondria due to increased fusion. This type of OPA1 overexpression did not significantly affect the health and reproduction of the animals, similar to the model in which moderate OPA1 overexpression was achieved by X-chromosome targeted OPA1 transgenesis [32–34]. Western blot analysis showed that the expression of OPA1 is significantly increased in the transgenic animals and the transgenic origin of the protein was verified with FLAG antibody (Fig 9) The normal amount of VDAC correlating with the mitochondrial quantity proved that the higher OPA1 level is not a consequence of an increased mitochondrial mass.

The mutant mice were not visibly different from the wild-type littermates, no changes were observed in the gravimetric parameters, blood pressure remained normal, and no arrhythmias occurred (Table 1). However echocardiographic examination showed a significantly decreased ejection fraction and trend-like increase of E/E' in transgenic mice during aging (Table 2). The deterioration of systolic function was not associated with fibrotic remodeling or cardiomyocyte diameter changes (Figs 2 and 3) and neither BNP level, nor the wet-to-dry lung ratio did not suggest heart failure (Table 1). The electron microscopic ultrastructural examination of the interfibrillar mitochondria showed a significant difference in mitochondrial size between the two groups. The mitochondria of transgenic mice were more heterogeneous in size and their mean area was larger. We performed a mitochondrial network analysis; however, the differences in mitochondrial size observed in the ultrastructural analysis were not visible at the light microscopic level.

A more significant difference was observed in the inner structure of mitochondria (Fig 5). The vacuole-like intercristal widenings/expansions and the disruption of cristae are presumably due to the deposition of large amount of transgenic protein. The fact that OMA1 is unable to cleave these proteins may enhance the mitochondrial dysfunction. This inner membrane instability may also explain the respiratory capacity of the transgenic cardiomyocytes (Fig 9), as we detected a significant decrease in membrane potential in the transgenic groups (Fig 8).

We hypothesize that OPA1 overproduction resulted the promotion of fusion processes which is evident in the increase of mitochondrial size. Nevertheless, no changes were observed in mitochondrial mass, including mitochondrial DNA copy number and VDAC protein levels (Figs 10 and 11). We expected that the L-OPA protein quantity would increase in the transgenic animals. However, Western blot examination showed that the amount of both forms increased and at least partly as we expected the L-OPA1/S-OPA1 ratio shifted towards the long form (Fig 9A–9C). The level of OMA1 protease involved in OPA1 processing did not change and the level of YME1L1 decreased in transgenic mice (Fig 9G–9H). However, changes in the OMA1/YME1L1 ratio may also contribute to mitochondrial changes and cardiac dysfunction [54].

To explain the decreased ejection fraction, we must consider the abnormalities observed in the ultrastructure of the mitochondria, possibly due to the excess amount of transgenic protein. This may lead to a consequential decrease in the effectiveness of mitochondrial respiration as well. The processes of mitophagy are present in the transgenic strain, as indicated by the elevated levels of Pink/Parkin proteins. However, the low levels of the pro-apoptotic BNIP3 do not suggest severe mitophagy and proteosomal degradation.

The processes of mitophagy are present in the transgenic strain, as indicated by the elevated levels of PINK/Parkin proteins. Additionally, a higher fraction of mitochondria in the smaller size ranges was observed in the transgenic group during electron microscopic evaluations. This tendency may indicate normal levels of fragmentation associated with mitophagy, as fragmented mitochondria were not observed at the light microscopy level. Therefore, we do not observe increased fragmentation but rather stronger mitophagy control.

There is ample data suggesting that promoting fusion processes has beneficial effects and enhances cell viability and the elevated level of OPA1 protein stabilize the membrane structure. Moreover as we mentioned above S-OPA1 possesses the ability to preserve cristae structure and maintain energetic functions of the mitochondria and the production of S-OPA1 via OPA1 cleavage may serve as a protective mechanism in cells under stress conditions [51–57].

However, in our model, the increased quantity of OPA1 may also be counterproductive concerning mitochondrial ultrastructural abnormalities and the shift in the balance of OMA1/YME1L1 ratio that have led to consequent deterioration of mitochondrial function and as a

result of these gradually accumulating harmful changes, a decreased ejection fraction has developed in older transgenic animals [54].

Despite of the ultrastructural alterations observed in this chronic model, due to the membrane stabilizing role of OPA1, it deserves further consideration to investigate the possible advantages/benefits of this genotype.

We should consider the possibility that the adverse phenomena experienced at this stage might be counteracted by some undiscovered beneficial properties of this model, which could become apparent in acute stress scenarios.

## Supporting information

**S1 Fig. Supplementary information for Fig 2.**
(PDF)

**S2 Fig. Supplementary information for Fig 3.**
(PDF)

**S3 Fig. Supplementary information for Fig 4.**
(PDF)

**S4 Fig. Supplementary information for Fig 5.**
(PDF)

**S5 Fig. Supplementary information for Fig 7.**
(PDF)

**S6 Fig. Supplementary information for Fig 8.**
(PDF)

**S7 Fig. Supplementary information for Fig 9.**
(PDF)

**S8 Fig. Supplementary information for Fig 10.**
(PDF)

**S9 Fig. Supplementary information for Fig 11.**
(PDF)

**S10 Fig. Supplementary information for Fig 12.**
(PDF)

**S11 Fig. Supplementary information for WB.**
(PDF)

## Author Contributions

**Conceptualization:** Ferenc Gallyas, Kalman Toth, Robert Halmosi, Laszlo Deres.

**Data curation:** Kitti Bruszt, Orsolya Horvath, Katalin Ordog, Szilard Toth, Laszlo Deres.

**Investigation:** Kitti Bruszt, Orsolya Horvath, Kata Juhasz, Eszter Vamos, Katalin Fekete.

**Methodology:** Kitti Bruszt, Katalin Ordog, Kata Juhasz, Eszter Vamos, Katalin Fekete.

**Project administration:** Orsolya Horvath, Katalin Ordog, Katalin Fekete, Laszlo Deres.

**Resources:** Robert Halmosi.

**Supervision:** Ferenc Gallyas, Kalman Toth, Robert Halmosi, Laszlo Deres.

**Validation:** Szilard Toth.

**Visualization:** Szilard Toth, Laszlo Deres.

**Writing – original draft:** Kitti Bruszt.

**Writing – review & editing:** Szilard Toth, Ferenc Gallyas, Kalman Toth, Robert Halmosi, Laszlo Deres.

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
