## [Decision Letter · Decision Letter 0]

27 May 2024

PONE-D-24-07670Cardiac effect of OPA1 overexpression in micePLOS ONE

Dear Dr. Deres,

Thank you for submitting your manuscript to PLOS ONE. After careful consideration, we feel that it has merit but does not fully meet PLOS ONE’s publication criteria as it currently stands. Therefore, we invite you to submit a revised version of the manuscript that addresses the points raised during the review process.

The reviewer submitted his/her report. As you can see, although the merit of your work, several concerns have been raised by the reviewer, with whom I substantially agree. My decision is that major revision are needed before the paper could be accepted for publication in PLOS One.

We look forward to receiving your revised manuscript.

Kind regards,

Enrico Baruffini, Ph.D.

Academic Editor

PLOS ONE

Journal Requirements:

2. To comply with PLOS ONE submissions requirements, in your Methods section, please provide additional information regarding the experiments involving animals and ensure you have included details on (1) methods of sacrifice and (2) efforts to alleviate suffering.

- https://aok.pte.hu/biotechnologia-msc?

In your revision ensure you cite all your sources (including your own works), and quote or rephrase any duplicated text outside the methods section. Further consideration is dependent on these concerns being addressed.

5. Thank you for stating the following financial disclosure: "This research was funded by the National Research, Development, and Innovation Office in Hungary within the framework of the project TKP2021-EGA-17."  

7. PLOS ONE now requires that authors provide the original uncropped and unadjusted images underlying all blot or gel results reported in a submission’s figures or Supporting Information files. This policy and the journal’s other requirements for blot/gel reporting and figure preparation are described in detail at https://journals.plos.org/plosone/s/figures#loc-blot-and-gel-reporting-requirements and https://journals.plos.org/plosone/s/figures#loc-preparing-figures-from-image-files. When you submit your revised manuscript, please ensure that your figures adhere fully to these guidelines and provide the original underlying images for all blot or gel data reported in your submission. See the following link for instructions on providing the original image data: https://journals.plos.org/plosone/s/figures#loc-original-images-for-blots-and-gels.   

Reviewers' comments:

Reviewer's Responses to Questions

**Comments to the Author**

1. Is the manuscript technically sound, and do the data support the conclusions?

Reviewer #1: Partly

2. Has the statistical analysis been performed appropriately and rigorously? 

Reviewer #1: Yes

3. Have the authors made all data underlying the findings in their manuscript fully available?

Reviewer #1: Yes

4. Is the manuscript presented in an intelligible fashion and written in standard English?

Reviewer #1: Yes

5. Review Comments to the Author

Reviewer #1: The article entitled ‘Cardiac effect of OPA1 overexpression in mice” by Bruszt K. et al. reports various observations on the effects of overexpression of the mitochondrial fusion protein OPA1 in the heart of mouse.

Overall, the manuscript is written in a clear way, although there are several typos throughout the text. I think that the abstract should be completely re-written, as it does not reflect the content of the article in terms of scientific results and it ends on a somehow subjective question, which is very unusual.

I have major concerns about this manuscript.

1) The title is misleading, as the authors do not really explore cardiac OPA1 overexpression, rather overexpression of a variant not cleavable by OMA1 (delta S1) in addition to normal OPA1 content. Cardiac OPA1 overexpression would mean that the transgene is a wild-type coding sequence of OPA1.

2) The delta S1 transgene cannot be cleaved by OMA1, but it can be cleaved by YME1L1 at cleaving site S2. So, I do not really understand the strategy, because this only prevents to cleave OPA1 under stress conditions (OMA1), but S-OPA1 can still be generated.

3) Overall, the histology experiments lack positive controls. For the interstitial collagen deposition experiment with Picrosirius red, the authors should include positive control samples that display collagen deposition (for example, heart from old mouse, i.e. more than 12 months) to show that the staining is working. Indeed, picrosirius red staining usually shows collagen fibrils in red on a pale-yellow background, which facilitates their quantification. Here the picture shows a homogenous staining where such fibrils are hardly visible.

4) For the diameter of cardiomyocytes, I think that using the picrosirius red picture to estimate the area of cardiomyocytes is imprecise. The authors should use a fluorescence staining, for example with laminin alpha 2, to delineate each cardiomyocyte and estimate the area.

5) The authors describe alterations of mitochondrial cristae structure in transgenic hearts, but on the pictures provided, it is not visible. Please provide enlarged inserts showing these alterations.

6) The mitochondrial network evaluation on NMCM is very subjective, as the authors state “a slight fragmentation of the mitochondrial network structure was observed, but the degree of this change was not significant.” The authors cannot conclude that there is no significant change, without thorough evaluation of the pictures. Please provide quantification analysis of the various mitochondrial shapes (tubular, fragmented, intermediate, etc…) on the NMCM pictures to bring a firm conclusion to this experiment.

7) The authors used JC-1 for evaluation of the mitochondrial membrane potential, but this dye has pitfalls (see Perry et al., 2011 doi: 10.2144/000113610). The authors should perform new experiments with TMRM or TMRE, with calibration with oligomycin and FCCP. Alternatively, they can use a double staining with TMRM (dependent of mitochondrial membrane potential) and Mitotracker green (independent of mitochondrial membrane potential), to show the proportion of active mitochondria in both populations.

8) The western blot data are also confusing. The authors mention an increase in S-OPA1 isoform in the transgenic group, but there are no quantifications of L-OPA1 and S-OPA1 alone, just a global OPA1 quantification. Also, the authors state page 12 that the L-OPA1/S-OPA1 ratio is significantly lower in transgenic mice, while the figure 9C shows the opposite! I am very surprised by the low OPA1 expression in Wild-Type mice. Given the importance of this protein for this article, I would recommend testing a second antibody to confirm this. The authors use VDAC expression as a reflection of mitochondrial mass, but this is only one parameter to estimate mitochondrial biogenesis. To infer this, the authors would need to evaluate mtDNA copy number, Citrate synthase activity, cardiolipin content, etc…Also, some of the blots are not convincing, in particular for PARKIN and PINK1, which show variability and low expression in the WT animals, with only 4 samples of each genotype. I would recommend repeating the blots at least for these 2 proteins with more samples.

For the above-mentioned reasons, my recommendation for this article is Major revisions.

6. PLOS authors have the option to publish the peer review history of their article (what does this mean?). If published, this will include your full peer review and any attached files.

Reviewer #1: No

---

## [Author Response · Author response to Decision Letter 0]

5 Aug 2024

Response to Reviewer Comments

Thank you for taking the time to read our manuscript and for providing valuable feedback. We are sending hereby our responses to your comments. We have introduced the appropriate changes and additive measurements in our resubmitted manuscript.

Point 1. Abstract and title of the article

We agree with the suggestion regarding the abstract. We have re-written the abstract to better reflect the scientific results presented in the article. Additionally, we have changed the title of the article as the Reviewer suggested.

Point 2. The delta S1 transgene cannot be cleaved by OMA1, but it can be cleaved by YME1L1, I do not really understand the strategy, because this only prevents to cleave OPA1 under stress conditions (OMA1), but S- OPA1 can still be generated.

Expression of optic atrophy 1 (OPA1), a mitochondrial fusion protein, was decreased in both human and rat HF. OPA1 is important for maintaining normal cristae structure and function, for preserving the inner membrane structure and for protecting cells from apoptosis. 

Reduction of OPA1 expression with siRNA resulted in increased apoptosis and fragmentation of the mitochondria. 

Overexpression of OPA1 increased mitochondrial tubularity but did not protect against simulated ischaemia-induced apoptosis. Cytochrome c release from the mitochondria was increased both with reduction in OPA1 and with overexpression of OPA1. Mouse tissues express four isoforms: 1, 5, 7, and 8 (Akepati et al., 2008). Isoforms 1 and 7 generate a combination of L-OPA1 and one or two forms of S-OPA1, respectively. 

Our strategies based on that the regulation of processes mediated by the OPA1 protein also involves fine-tuning the L-OPA1/S-OPA1 ratio, as these two forms perform different functions. The OMA1 protease is activated by stress, mitochondrial damage and membrane potential reduction and this protein is responsible for cleaving OPA1 at the S1 site. YME1L1 constitutively regulates OPA1 by cleavage at the S2 site. 

It has been proposed that stress-triggered cleavage of OPA1 leads to fragmentation of mitochondria and enhances cell sensitivity to death. However, a recent research has revealed that S-OPA1 possesses the ability to preserve cristae structure and maintain energetic functions due to its GTPase activity, despite its lack of fusion capability. These data indicates that the production of S-OPA1 via OPA1 cleavage may serve as a protective mechanism in cells under stress conditions

We suggest that deleting the OMA1 S1 cleavage site prevents the stress response-induced S-OPA1 dominance and consequent mitochondrial fragmentation. The stable L-OPA1 contributes to cristae structure stability, while the constitutively active YME1L1 maintains the necessary S-OPA1 levels for normal function, including normal fission induction and chaperone functions.

We are grateful for the constructive criticism and have clarified our approach in the introduction and discussion section.

Point 3. Overall, the histology experiments lack positive controls. For the interstitial collagen deposition experiment with Picrosirius red, the authors should include positive control samples that display collagen deposition (for example, heart from old mouse, i.e. more than 12 months) to show that the staining is working. Indeed, picrosirius red staining usually shows collagen fibrils in red on a pale-yellow background, which facilitates their quantification. Here the picture shows a homogenous staining where such fibrils are hardly visible.

We agree that including positive control samples is crucial for validating the histology experiments. In response, we have repeated the staining using Masson's trichrome stain with the recommended positive control samples from 15-month-old mice. The results are indeed more convincing with this method. However, despite these improvements, we still did not observe a significant change in the extent of fibrosis. Thank you for your valuable suggestion.

Point 4. For the diameter of cardiomyocytes, I think that using the picrosirius red picture to estimate the area of cardiomyocytes is imprecise. The authors should use a fluorescence staining, for example with laminin alpha 2, to delineate each cardiomyocyte and estimate the area.

The Reviewer’s point about the precision of using picrosirius red to estimate the area of cardiomyocytes is valid. Using fluorescence staining with laminin alpha 2 would indeed be more accurate, as it delineates each cardiomyocyte more clearly. While classic histochemical stains have been extensively used for such evaluations in the past and even today, we did not find it necessary to perform immunohistochemistry since neither gravimetry nor echocardiographic parameters indicated hypertrophic remodeling.

However, we would have been willing to repeat the staining as suggested by the Reviewer. Unfortunately, our samples are formaldehyde-fixed and paraffin-embedded, which is not optimal due to autofluorescence issues. If the Reviewer insists, we are willing to declare that the measurement is imprecise and approximate in a Limitations section. We hope this minor issue does not detract from the overall value of our findings.

Point 5. The authors describe alterations of mitochondrial cristae structure in transgenic hearts, but on the pictures provided, it is not visible. Please provide enlarged inserts showing these alterations. 

Thank you for your comment. Upon closer examination, we observed smaller vacuole-like intercristal widenings and minor disruptions of the mitochondrial cristae structure in the transgenic group. Subjectively, the two groups can be clearly distinguished. By using a blinded method to analyze the images at a magnification of 12,000-30,000x, which is approximately the maximum capability of our available system, these ultrastructural differences became evident. To clarify these differences, we have indicated them with arrowheads.

Point 6. The mitochondrial network evaluation on NMCM is very subjective, as the authors state “a slight fragmentation of the mitochondrial network structure was observed, but the degree of this change was not significant.” The authors cannot conclude that there is no significant change, without thorough evaluation of the pictures. Please provide quantification analysis of the various mitochondrial shapes on the NMCM pictures to bring a firm conclusion to this experiment.

Thank you for your comment. We acknowledge the subjectivity in evaluating the mitochondrial network on NMCMs. To draw a firm conclusion, we conducted a thorough quantification analysis of the mitochondrial shapes using the Mitochondria Network Analysis tool (MiNA) on the ImageJ interface. 

For this analysis, images were imported into ImageJ and processed into 8-bit grayscale images. To enhance the quality, we applied an unsharp mask, enhanced local contrast, and performed median filtering. The images were then binarized to generate black foreground mitochondria images against a white background and converted to a skeleton that represents the features of the original mitochondria image in the form of lines. The skeletons were further analyzed using the ImageJ "analyze skeleton" plugin, which measures the length of each branch.

In total, 10 cells from each group were measured and compared using an unpaired t-test. This quantification analysis provides a more objective basis for our conclusions regarding the mitochondrial network structure.

Based on the results, we have reviewed, corrected, and refined our conclusions regarding fragmentation. We appreciate the constructive suggestion.

Point 7. The authors used JC-1 for evaluation of the mitochondrial membrane potential, but this dye has pitfalls (see Perry et al., 2011 doi: 10.2144/000113610). The authors should perform new experiments with TMRM or TMRE, with calibration with oligomycin and FCCP. Alternatively, they can use a double staining with TMRM (dependent of mitochondrial membrane potential) and Mitotracker green (independent of mitochondrial membrane potential), to show the proportion of active mitochondria in both populations.

Thank you for your suggestion. We acknowledge that our experimental protocol using JC-1 indeed carried potential pitfalls. The article cited by the reviewer does highlight the limitations of the JC-1 dye (Perry et al., 2011, doi: 10.2144/000113610). Specifically, it mentions that JC-1 labeling is sensitive to residual H2O2 in the experimental system, which is irrelevant in our case, and that the equilibration time is closely linked to surface-to-volume (S/V) ratios. This means that in cell populations with significant heterogeneity in S/V ratios, such as neurons with extensive processes, this could be problematic.

Another methodological concern raised is that while the monomer (green) form of JC-1 equilibrates on a timescale of approximately 15 minutes, the aggregate (red) form—necessary for using JC-1 as a ratiometric probe for Δψm—takes about 90 minutes to equilibrate in cardiomyocytes (Mathur et al., 2000, doi: 10.1016/s0008-6363(00)00002-x). The study concludes that under these incubation conditions and at a concentration of at least 1 μM, JC-1 is the optimal dye for measuring Δψm in cardiomyocytes.

In response to these considerations, we redesigned our experiment by extending the incubation time and introducing FCCP as a pharmacological control to confirm that directional changes in the dye signal are appropriately interpreted. Based on the results obtained, the mitochondrial membrane potential was found to be significantly lower in the transgenic group. We have incorporated and concluded these findings in the manuscript.

Thank you for the constructive suggestion and your valuable input.

Point 8. The western blot data are also confusing. The authors mention an increase in S-OPA1 isoform in the transgenic group, but there are no quantifications of L-OPA1 and S-OPA1 alone, just a global OPA1 quantification. Also, the authors state page 12 that the L-OPA1/S-OPA1 ratio is significantly lower in transgenic mice, while the figure 9C shows the opposite! I am very surprised by the low OPA1 expression in Wild-Type mice. Given the importance of this protein for this article, I would recommend testing a second antibody to confirm this. The authors use VDAC expression as a reflection of mitochondrial mass, but this is only one parameter to estimate mitochondrial biogenesis. To infer this, the authors would need to evaluate mtDNA copy number, Citrate synthase activity, cardiolipin content, etc...Also, some of the blots are not convincing, in particular for PARKIN and PINK1, which show variability and low expression in the WT animals, with only 4 samples of each genotype. I would recommend repeating the blots at least for these 2 proteins with more samples. 

Thank you for your comment. We apologize for the confusion regarding the Western blot data. The L-OPA1/S-OPA1 ratio is significantly higher in the transgenic mice, as correctly shown in Figure 9 C. We have corrected this mistake in the text. 

We used the monoclonal OPA1 Cell Signaling #80471 antibody, which is highly specific for both L-OPA1 and S-OPA1. This is visible in two bands at 100 kDa and 80 kDa. We quantified these separately, but only visualized the L-OPA1/S-OPA1 ratio to decrease redundancies, as we felt that the given figure already contained a lot of data and this approach better fit the logic of the manuscript. If the reviewer insists, we can replace it with the direct measurement results. 

The low level of OPA1 expression in wild-type mice is only apparent. Western blotting is a semi-quantitative technique, and our gel documentation system uses auto-exposure, which accumulates data relative to the strongest band to prevent overexposure and pixel saturation make the densitometric measurements more precise. Following your suggestion, we repeated the measurements using a different antibody (Invitrogen Opa1 1E8-1D9) and fine-tuned the exposure time to ensure more accurate detection of OPA1 levels. 

We acknowledge that using VDAC expression as a reflection of mitochondrial mass is only one parameter to estimate mitochondrial biogenesis. In response to your suggestion, we have purchased the NovaQUANT™ Mouse Mitochondrial to Nuclear DNA Ratio Kit and performed the additional measurements. This will provide a more comprehensive evaluation of mitochondrial biogenesis, including mtDNA copy number.

The variability in the expression of PARKIN and PINK1 in the WT animals is indeed high, which is unfortunately common with in vivo samples. We have repeated the measurements using freshly prepared samples, and the results have been incorporated into the manuscript. Additionally, after the repeated measurements, the changes in both PINK and PARKIN proteins proved to be significant.

We would like to express our sincere gratitude for the detailed and helpful comments, which have significantly contributed to the improvement of our manuscript.

---

## [Decision Letter · Decision Letter 1]

2 Sep 2024

Cardiac effects of OPA1 protein promotion in a transgenic animal model

PONE-D-24-07670R1

Dear Dr. Deres,

We’re pleased to inform you that your manuscript has been judged scientifically suitable for publication and will be formally accepted for publication once it meets all outstanding technical requirements.

Kind regards,

Enrico Baruffini, Ph.D.

Academic Editor

PLOS ONE

Additional Editor Comments (optional):

The reviewer endorsed your paper, since his/her comments have been addressed.

The paper can now be accepted by PLoS ONE.

Reviewers' comments:

Reviewer's Responses to Questions

**Comments to the Author**

1. If the authors have adequately addressed your comments raised in a previous round of review and you feel that this manuscript is now acceptable for publication, you may indicate that here to bypass the “Comments to the Author” section, enter your conflict of interest statement in the “Confidential to Editor” section, and submit your "Accept" recommendation.

Reviewer #1: All comments have been addressed

2. Is the manuscript technically sound, and do the data support the conclusions?

Reviewer #1: (No Response)

3. Has the statistical analysis been performed appropriately and rigorously? 

Reviewer #1: (No Response)

4. Have the authors made all data underlying the findings in their manuscript fully available?

Reviewer #1: (No Response)

5. Is the manuscript presented in an intelligible fashion and written in standard English?

Reviewer #1: (No Response)

6. Review Comments to the Author

Reviewer #1: (No Response)

7. PLOS authors have the option to publish the peer review history of their article (what does this mean?). If published, this will include your full peer review and any attached files.

Reviewer #1: No

---

## [Editor Report · Acceptance letter]

11 Sep 2024

PONE-D-24-07670R1 

PLOS ONE

Dear Dr. Deres, 

I'm pleased to inform you that your manuscript has been deemed suitable for publication in PLOS ONE. Congratulations! Your manuscript is now being handed over to our production team.

Kind regards, 

on behalf of

Dr. Enrico Baruffini 

Academic Editor

PLOS ONE